# Anserine and Carnosine Induce HSP70-Dependent H_2_S Formation in Endothelial Cells and Murine Kidney

**DOI:** 10.3390/antiox12010066

**Published:** 2022-12-28

**Authors:** Charlotte Wetzel, Tilman Pfeffer, Ruben Bulkescher, Johanna Zemva, Sergio Modafferi, Alessandra Polimeni, Angela Trovato Salinaro, Vittorio Calabrese, Claus Peter Schmitt, Verena Peters

**Affiliations:** 1Centre for Pediatric and Adolescent Medicine, University Hospital Heidelberg, 69120 Heidelberg, Germany; 2Department of Medicine I and Clinical Chemistry, Heidelberg University Hospital, 69120 Heidelberg, Germany; 3Department of Biomedical and Biotechnological Sciences, University of Catania, 95125 Catania, Italy

**Keywords:** anserine, carnosine, diabetic nephropathy, hydrogen sulfide

## Abstract

Anserine and carnosine have nephroprotective actions; hydrogen sulfide (H_2_S) protects from ischemic tissue damage, and the underlying mechanisms are debated. In view of their common interaction with HSP70, we studied possible interactions of both dipeptides with H_2_S. H_2_S formation was measured in human proximal tubular epithelial cells (HK-2); three endothelial cell lines (HUVEC, HUAEC, MCEC); and in renal murine tissue of wild-type (WT), carnosinase-1 knockout *(Cndp1*-KO) and *Hsp70*-KO mice. Diabetes was induced by streptozocin. Incubation with carnosine increased H_2_S synthesis capacity in tubular cells, as well as with anserine in all three endothelial cell lines. H_2_S dose-dependently reduced anserine/carnosine degradation rate by serum and recombinant carnosinase-1 (CN1). Endothelial *Hsp70*-KO reduced H_2_S formation and abolished the stimulation by anserine and could be restored by *Hsp70* transfection. In female *Hsp70*-KO mice, kidney H_2_S formation was halved. In *Cndp1*-KO mice, kidney anserine concentrations were several-fold and sex-specifically increased. Kidney H_2_S formation capacity was increased 2–3-fold in female mice and correlated with anserine and carnosine concentrations. In diabetic *Cndp1*-KO mice, renal anserine and carnosine concentrations as well as H_2_S formation capacity were markedly reduced compared to non-diabetic *Cndp1*-KO littermates. Anserine and carnosine induce H_2_S formation in a cell-type and Hsp70-specific manner within a positive feedback loop with CN1.

## 1. Introduction

The histidine-containing dipeptide carnosine is synthesized in muscle and kidney by carnosine synthase and is methylated to anserine via carnosine methyltransferase [1]. Both dipeptides are degraded by carnosinase 1 (CN1), a product of the *Cndp1* gene [2]. Diabetic patients with *Cndp1* gene variants, which are associated with a lower serum CN1 activity, have a lower risk of nephropathy [3]. Serum CN1 concentrations have been correlated with renal fibrosis, oxidative stress and tubular injury [4]. In rodent models of diabetes type 1 and 2, carnosine treatment reduced oxidative stress, carbonyl stress and advanced glycation end-product (AGE) formation, and improved glucose homeostasis, all of which are associated with less structural and functional renal damage [5,6]. In contrast to rodents, anserine and carnosine are rapidly metabolized by serum CN1 in humans; still, the first clinical trials yielded some positive results [7,8], possibly by carnosine delivery to the kidney via erythrocytes [9]. Carnosine has been approved as a nutritional supplement and is well tolerated [10]. The mode of action of anserine and carnosine, however, is not fully understood. Anti-inflammatory, antioxidant, anti-glycation and reactive carbonyl-quenching properties have been reported [11,12]. Depending on the experimental setting, quenching of highly toxic methylglyoxal is low [12,13].

Carnosine induces heat shock protein (HSP) 70 in podocytes, also in the presence of metabolic stress [14], and anserine induces HSP70 in human proximal tubular epithelial cells (HK-2) cells together with reactive metabolite stress [15]. HSP70 mediates a variety of cellular functions, prominently including protein folding and regulatory processes of cell repair [16,17]. HSP70 expression is altered in diabetic patients, and in clinical trials, circulating levels of HSP70 were associated with diabetic complications [18]. HSP70 increases in response to hydrogen sulfide (H_2_S) [19], which has a broad spectrum of biological activities, including antioxidant, cytoprotective, vasodilatory, anti-inflammatory and antifibrotic effects [17,20]. H_2_S is primarily synthesized from homocysteine and L-cysteine by cystathionine-β-synthase (CBS) and cystathionine-γ-lyase (CSE) [21]. CBS is the major H_2_S-producing enzyme in brain, while CSE is abundant in the mammalian cardiovascular and respiratory system and possibly the main H_2_S-forming enzyme in the liver, kidney and pancreas [22]. Lower H_2_S levels were observed in many renal pathologies, including diabetic nephropathy [20,23], and plasma H_2_S levels correlate with glomerular filtration rate in patients with chronic kidney disease [24]. In animal models of renal disorders, H_2_S donors restored H_2_S levels and improved renal function [25,26]. H_2_S donors such as thiosulfate are well-tolerated, and their therapeutic potential is currently studied in clinical trials [27,28,29,30]. 

Considering the protective effects of anserine and carnosine and of H_2_S and their common interaction with HSP70, we studied possible interactions of the two dipeptides with H_2_S synthesis. Anserine and carnosine increase H_2_S in a cell-specific, HSP70-dependent mode, findings which shed novel light on current therapeutic approaches involving histidine-containing dipeptides and H_2_S donors.

## 2. Materials and Methods

### 2.1. Cell Culture

Immortalized human proximal tubular epithelial cells (HK-2, American Type Culture Collection CRL-2190, Manassas, VA, USA) were grown in RPMI 1640 Glutamax Medium (ThermoFisher scientific, Waltham, MA, USA) with 10% fetal calf serum (ThermoFisher scientific, Waltham, MA, USA) (*v*/*v*), 1% penicillin and streptomycin (ThermoFisher scientific, Waltham, MA, USA) (*v*/*v*) at 37 °C and 5% CO_2_. Immortalized murine cardiac endothelial cells (MCEC, CLU510 Cedarlane Laboratories, Burlington, ON, Canada) were grown in Dulbecco’s Modified Eagle Medium (DMEM) (ThermoFisher scientific, Waltham, MA, USA) with 5% fetal calf serum (ThermoFisher scientific, Waltham, MA, USA) (*v*/*v*), 1% penicillin and streptomycin (*v*/*v*), 1% Ampicillin B (*v*/*v*) and 1% HEPES (ThermoFisher Scientific, Waltham, MA, USA) (*v*/*v*) at 37 °C and 5% CO_2_. Primary human umbilical vein endothelial cells (HUVEC, PromoCell, Heidelberg, GER) and primary human umbilical artery endothelial cells (HUAEC, PromoCell, Heidelberg, GER) were grown in Endothelial Cell Growth Medium with SupplementMix (PromoCell, Heidelberg, GER) and 1% penicillin and streptomycin (*v*/*v*) at 37 °C and 5% CO_2_. Cells were split by using 0.25% EDTA trypsin (Thermo Fisher Scientific, Waltham, MA, USA) for dissociation. Medium LDH measurements excluded anserine- and carnosine-induced cytotoxicity, and the applied concentrations were below the previously published EC 50 [13].

### 2.2. Hsp70-Knockout MCEC

*Hsp70*-knockout was generated by transfection (Neon Transfection System, Invitrogen, Waltham, MA, USA) of 10^6^ with a vector from Sigma-Aldrich, targeting the stress-inducible Hsp70 variant Hspa1a (Gene ID: 193740; targeting sequence of the gRNA: TGTGCTCAGACCTGTTCCG). The vector contained the respective gRNA target sequences, the Cas9 endonuclease gene and a fluorescent reporter gene (GFP for *Hspa1a*), that was used for single cell isolation by FACS. Clones were cultured, and genome, mRNA and protein analysis were performed to confirm successful knockout of *Hspa1a.*

### 2.3. Maximal H_2_S Production Capacity

H_2_S was identified by detection of Ag_2_S according to Ahn et al. [31]. Plates were coated with AgNO_3_/Nafion/PVP solution and dried at room temperature (1 h) before incubation start. Cells were seeded on 96-well plates (2 × 10^4^ per well for HK-2, 5 × 10^4^ per well for HUAEC, HUVEC and MCEC) followed by a 22 h growth period and treated with L-homocysteine, anserine and carnosine for 48 h; L-homocysteine was renewed after 24 h. Incubation with a CBS inhibitor (O-(Carboxymethyl)-hydroxylamin-hemihydrochlorid; AOAA) and a CSE inhibitor (DL-proparglycine; PAG) for 48 h was used to demonstrate the involvement of those enzymes in H_2_S production capacity.

Murine kidney tissue (20 mg in protease-inhibited DPBS buffer) was homogenized and centrifuged at 4 °C and 10,000× *g* for 10 min. Supernatant was treated for 6 h with L-cysteine.

Absorbance was measured at 310 nm, and Ag_2_S production was normalized to protein concentration (DC™ Protein Assay; Bio-Rad Laboratories, Hercules, CA, USA).

### 2.4. Anserine and Carnosine Concentrations

Anserine and carnosine concentrations in kidney tissue of WT, *Cndp1*-KO and *Hsp70*-KO mice were measured fluorometrically using HPLC as described previously [32]. Next, 50 mg of frozen renal tissue was homogenized, diluted and subsequently derivatized with carbazole-9-carbonyl chloride (CFC) and measured by fluorescence detection (RF-20A, Shimadzu, Kyoto, Japan; Jupiter column C18, 300 Å, 5 µm particle size, 250 × 4.6 mm, Phenomenex, Aschaffenburg, Germany). Mobile phase consisted of a binary gradient with 82% solution A (50 mmol/L acetate buffer in distilled water; pH 4.37) and 18% solution B (acetonitrile, methanol and tetrahydrofuran 70:25:5 (*v*/*v*/*v*)).

### 2.5. Western Blotting

Cell samples were lysed in RIPA buffer (radio-immunoprecipitation assay buffer: 150 mM NaCl, 0.1% Triton X-100, 0.5% sodium deoxycholate, 0.1% SDS and 50 mM Tris-HCl; pH 8.0) and protease inhibitor (cOmplete tablets, Mini EASYpack, Roche Diagnostics, Mannheim, Germany) and separated by SDS-PAGE in 8% polyacrylamide gels. Samples were transferred to a nitrocellulose membrane by semi-dry blot. The membrane was then blocked with 5% milk (1 h at room temperature) and incubated with anti-Hsp70-antibody (HSP70 Polyclonal antibody, ProteinTech, Rosemont, IL, USA, 1:10,000 in 5% milk; 1.5 h at room temperature). After washing with Tris-buffered saline with Tween 20 (TBS-T), the membrane was incubated with a secondary HRP-conjugated antibody (1:1000 in 5% milk) for 1 h at room temperature. Protein expression of the target protein was normalized to β-Actin expression of the representative sample.

Western blots were developed with Clarity Western ECL Substrate (Biorad, Hercules, CA, USA), imaged via a fluorescence imaging system (PEQLAB fusion, PEQLAB, Erlangen, Germany) and quantified via ImageJ program (NIH).

### 2.6. Carnosinase Activity

CN1 activity was assayed according to the method described by Teufel et al. (2003) [2]. The reaction was initiated by addition of carnosine to human serum carnosinase or recombinant enzyme (rCN1; R&D Systems, Minneapolis, MN, USA) at pH of 7. The reaction was stopped after defined periods by adding 1% trichloracetic acid (final concentration in the test 0.3%). Liberated histidine was derivatized by adding o-phtaldialdehyde (OPA), and fluorescence was read using a plate reader (MicroTek International Inc., Hsinchu, Taiwan).

### 2.7. Mice

Global *Cndp1-*KO mice were used for the performed experiments (strain C57B6/6J; 129S5-Cndp1tm1Lex; #032215; MMRRC UC, Davis, CA, USA; https://mmrrc.ucdavis.edu/protocols/032215Geno_Protocol.pdf (accessed on 1 July 2018), and global *Hsp70*-KO mice (strain C57B6/6N; 129S7-Hspa1/Hspa1btm1Dix; #000371-UCD; MMRRC UC, Davis, CA, USA) and their respective wild types (WT; strain C57B6/6J and C57B6/6N). Animals were housed at the Interfaculty Biomedical Facility (IBF) at Heidelberg University with a daily cycle of 12 h light and 12 h darkness at 24 °C as well as ad libitum supply of food and water. The global *Cndp1*-KO was confirmed by gene expression via PCR (PhireTM Tissue Direct PCR Master Mix, ThermoFisher Scientific). The global *Hsp70*-KO was confirmed by gene expression via genome-PCR, qPCR and Western blot analysis. Diabetes mellitus Type 1 (DMT1) was induced by 5–6 intraperitoneal injections of Streptozocin (STZ, 50 mg/kg BW) at the age of 9 to 10 weeks. After manifestation of DMT1, *Cndp1*-KO mice were kept for 12 weeks. Blood sugar levels were measured three times per week. Insulin (Lantus, Sanofi-Aventis, Frankfurt, Germany) was injected s.c. according to blood sugar levels. Blood glucose levels were maintained stable, and urinary ketones were not elevated at any time point (Strip tests, Macherey-Nagel, Düren, Germany). At sacrifice, organs were harvested and cryo-preserved. The studies were approved by the respective authorities (Regierungspräsidium Karlsruhe, Germany, 35-9165.81/G-209/16 and 35-9185.81/G-103/18).

### 2.8. Statistical Analysis

Data were obtained from at least three independent experiments and are given as mean ± standard deviation (SD). Statistical analysis was performed with GraphPad prism 9 using analysis of variance (ANOVA) with Tukey’s test. *p*-values of <0.05 were considered significant.

## 3. Results

### 3.1. Dipeptide-Induced H_2_S Formation in Tubular Epithelial and Endothelial Cells

Homocysteine dose-dependently increased H_2_S formation in HK-2 cells (Appendix A). At a homocysteine concentration of 5 mM, the H_2_S formation rate in HK-2, HUVEC, HUAEC and MCEC varied between 15 and 73 pmol H_2_S/mg protein (Figure 1, Appendix A). The addition of anserine dose-dependently increased H_2_S formation in HK-2 cells (Figure 1A, Appendix A); H_2_S formation from cysteine was below the detection limit in HK-2. Addition of 1 mM anserine increased H_2_S formation in all cell types, and the addition of 1 mM carnosine increased H_2_S formation in HK-2 cells only. Combined anserine and carnosine exposure had no additive effect beyond the anserine-mediated effects (Figure 1).

To study the interaction of H_2_S on anserine and carnosine availability, CN1 activity was measured at increasing concentrations of the H_2_S donor sodium disulfide (Na_2_S). Na_2_S dose-dependently reduced recombinant CN1 and human serum CN1 carnosine and anserine degradation activity (Figure 2, Appendix A).

### 3.2. Anserine-Induced H_2_S Formation Depends on HSP70

Since anserine, carnosine and H_2_S have previously been described to modulate Hsp70 expression, we studied H_2_S formation capacity in MCEC with *Hsp70*-KO. H_2_S formation was 60% lower in *Hsp70*-KO MCEC compared to WT MCEC. Anserine and carnosine both did not increase H_2_S formation in the *Hsp70*-KO MCEC (Figure 3A). Rescue of the *Hsp70*-KO by *Hsp70* transfection restored H_2_S formation to the level of WT MCEC (Figure 3A). Successful transfection of *Hsp70* could be demonstrated for two further vectors by detection of total protein concentration of HSPA1A via Western blot (Appendix A). In all three rescued cell clones, anserine increased H_2_S formation to a similar degree as in WT MCEC (Appendix A).

### 3.3. H_2_S Formation in Murine Kidney Tissue

We then studied ex vivo H_2_S formation capacity in kidney tissue of WT mice. Addition of homocysteine and cysteine at equimolar concentrations to kidney tissue homogenate both increased H_2_S formation, and homocysteine to a smaller extent (Appendix A). H_2_S formation was not influenced by five hours of fasting (Appendix A) and was markedly reduced by inhibition of both CBS and CSE activity, indicating that both enzymes are involved in kidney H_2_S formation (Appendix A). To demonstrate the role of renal HSP70 on H_2_S formation, kidney tissues of mice with a global Hsp70-KO were studied in 23- to 28-week-old animals. Ex vivo kidney H_2_S formation capacity was 50% lower in female Hsp70-KO kidneys compared to WT mice (Figure 3B). In male Hsp70-KO mice, H_2_S formation was within the range of WT littermates. Kidney anserine concentrations were similar in all four groups (Figure 3C, Appendix A).

### 3.4. Kidney Anserine Abundance and H_2_S Formation

Incubation of kidney tissue with anserine and carnosine for six hours did not increase H_2_S formation (Appendix A). We then studied the effect of a persistent increase of endogenous kidney anserine and carnosine concentrations on H_2_S formation in global *Cndp1*-KO mice. Male *Cndp1*-KO mice had 3-fold higher renal anserine concentrations, and female *Cndp1*-KO mice had 5- to 12-fold higher concentrations than respective WT controls, depending on age (Table 1).

Renal carnosine concentrations were below the detection limit in WT and markedly increased in *Cndp1*-KO mice. H_2_S formation was 2- to 3-fold higher in female *Cndp1*-KO mice compared to their respective WT controls, but not different between male *Cndp1*-KO and WT animals. H_2_S formation and anserine/carnosine concentrations were correlated in 23- to 25- and 47- to 51-week-old mice (Figure 4).

We then studied the impact of diabetes on kidney anserine and carnosine concentrations and H_2_S formation. Kidney anserine and carnosine concentrations were similar in type 1 diabetic and non-diabetic WT mice, and carnosine concentrations were below detection level. In diabetic *Cndp1*-KO mice, kidney anserine and carnosine concentrations, however, were lower compared to non-diabetic *Cndp1*-KO littermates and in diabetic mice not related with H_2_S formation (Table 1, Figure 5). In female diabetic *Cndp1*-KO mice, kidney H_2_S formation was reduced compared to female non-diabetic *Cndp1*-KO mice and similar to diabetic WT mice. In male diabetic *Cndp1*-KO mice, kidney H_2_S formation was even reduced compared to the male diabetic WT littermates.

## 4. Discussion

Anserine and carnosine are experimentally well-established compounds mitigating diabetic nephropathy, and the first clinical trials yielded beneficial effects [7,33,34,35]. Similarly, H_2_S improves outcome in acute and chronic kidney impairment, experimental sepsis, hemorrhagic shock and following ischemia/reperfusion [20]. The first clinical trials investigating the effects of the H_2_S donor Na_2_S_2_O_3_ in myocardial infarct in humans are ongoing [27].

Both anserine/carnosine and H_2_S act via multiple mechanisms, and activation of HSP70 is a prominent common pathway [14,15,19,36]. We now demonstrate the cell type-specific increase in H_2_S synthesis capacity by anserine and carnosine, mediated by HSP70 activation (Figure 6). Ex vivo studies of kidney tissue reconfirm a correlation of H_2_S synthesis with tissue anserine and carnosine concentrations and demonstrate the mechanistic role of HSP70 in kidney H_2_S synthesis.

There is growing evidence that histidine-containing dipeptides exert major protective functions in various disease states by interfering with specific pathomechanisms [17]. Carnosine acts as a carbonyl scavenger [12,37], an ion-chelating agent [10], as an Angiotensin-Converting Enzyme (ACE) inhibitor [38,39] and multifunctional antioxidant [40,41]. In HK-2 cells, anserine has a higher antioxidative capacity than carnosine [15]. These activities are mediated via hormetic processes involving Nrf2, Sirt-1, Trx, Hsp70, the glutathione system [12,17,41] and modulation of the nitric oxide formation and metabolism [42,43]. Anserine and carnosine increase HSP70 expression depending on the cell type. Carnosine increases HSP70 in podocytes [14], independent of glucose concentrations, but not in HK-2 cells. Anserine increases HSP70 in HK-2 cells in the presence of oxidative and glucose stress [14,15], and incubation with carnosine and anserine are well tolerated by cells [13]. Since H_2_S increases myocardial and cerebral HSP70 [19,34], we investigated a putative interaction of anserine and carnosine with H_2_S synthesis in vitro and demonstrated that carnosine dose-dependently increases H_2_S synthesis capacity in HK-2 cells, and anserine increases this capacity in capillary, aortic and umbilical vein endothelial cells, i.e., in endothelial cells derived from capillaries and large arterial and venous vasculature. The induction of H_2_S synthesis in endothelial cells by anserine was entirely HSP70-dependent. *Hsp70*-KO in capillary endothelial cells abolished the stimulatory effect of anserine on H_2_S formation, and transfection of *Hsp70* into *Hsp70*-KO cells restored it. In line with this, in *Hsp70*-KO mice, kidney H_2_S synthesis was reduced by 50%. In contrast, the carnosine-induced upregulation of H_2_S synthesis in HK-2 cells should be HSP70-independent, since carnosine does not activate *Hsp70* in HK-2 cells [15].

We then studied the impact of H_2_S within the highly regulated metabolism of anserine and carnosine via CN1. Carnosine is provided by nutrition and tissue carnosine synthase (*Carns1*), but the impact of the latter is uncertain. In *Carns1*-KO mice, brain and muscle are carnosine-deficient, and stores can be replenished by oral intake [44], but the *Carns1*-KO had no impact on kidney and brain markers of carbonyl and oxidative stress in healthy and diabetic mice [45]. In contrast, anserine and carnosine are rapidly degraded by CN1, and kidney tissue CN1 activity in healthy and in type 2 diabetic mice is correlated with kidney anserine and carnosine concentrations and the tissue carbonyl and oxidative stress level [46]. These findings suggest a gatekeeping role and protective function of CN1 for kidney histidine-containing dipeptide concentration. In line with this, diabetic patients, who are homozygous carriers of the *Cndp1* gene variant CTG5, have significantly lower serum CN1 concentrations and activity, and its concentrations independently predict eGFR [47]. Previous enzyme kinetic studies and molecular dynamic simulations revealed inhibition of carnosine degrading CN1 activity by competitive inhibition with anserine [48], and by thiol-containing compounds due to allosteric interactions [49]; the latter interaction might be the underlying mechanism of the CN1 inhibition by H_2_S. In the same direction, we now demonstrate dose-dependent CN1 inhibition by H_2_S donor Na_2_S, suggesting interaction of anserine and of H_2_S in a positive feedback loop, i.e., mutual reinforcement of both protective mechanisms.

To demonstrate the impact of the interaction of anserine and carnosine with H_2_S synthesis capacity in vivo, we studied *Cndp1*-KO mice. These mice exhibit a kidney-selective, age- and gender0dependent 2- to 9-fold increase in kidney tissue anserine and carnosine concentrations, and kidney function is unaltered [32]. The increased kidney anserine and carnosine concentrations could be reconfirmed, underlying molecular mechanisms of sex-specific differences, such as the influence of estrogens, have not yet been studied in detail. In these mice, kidney anserine and carnosine concentrations correlated with the kidney tissue H_2_S synthesis capacity, underpinning the significance of the mechanistic interactions demonstrated in vitro in the in vivo setting. Experimental studies demonstrated the upregulation of vascular endothelial cell H_2_S synthesis by estrogen [50]. H_2_S has been previously recognized as a toxic gas, but has emerged as an important gaseous signaling molecule, and administration of the H_2_S donor thiosulfate (2 × 15 g) is well tolerated by humans [51]. The action of H_2_S involves a variety of molecular mechanisms, such as activation of PI3K/Akt/eNOS pathway, suppressing ferroptosis or the antioxidant effect mediated by Nrf2 signaling [52,53,54]. In line with the in vitro findings of carnosine inducing H_2_S synthesis only in proximal tubular epithelial cells, but anserine in all cell lines studied, correlations of kidney H_2_S synthesis capacity were higher with tissue anserine than carnosine concentrations. Of note, short-term incubation of kidney tissue with anserine did not increase H_2_S synthesis capacity, suggesting slower anserine-induced actions than observed in vitro, but unspecific alterations in the ex vivo tissue homogenate devoid of blood supply cannot be excluded.

The impact of diabetes mellitus was demonstrated in type 1 diabetic *Cndp1*-KO mice. While kidney anserine and carnosine concentrations were in a similar range in diabetic and non-diabetic WT mice, concentrations of both histidine-containing dipeptides were 3-fold lower in the kidney of diabetic versus non-diabetic *Cndp1*-KO mice. In the diabetic mice, H_2_S formation rate was not correlated with the low kidney anserine and carnosine. Thus, the type 1 diabetes mellitus *Cndp1*-KO model does not allow for firm conclusions on their interaction. In type 2 diabetes db/db mice of similar age, kidney anserine and carnosine concentrations were also reduced [46]. Plasma and kidney H_2_S levels have repeatedly been reported to be low in diabetes mellitus [20], except for one study with double transgenic Balb/c mice sacrificed at the age of 13.5 weeks that had increased kidney H_2_S [55], a finding which could be reproduced here in male, but not in female STZ mice with a similar duration of diabetes. In female STZ *Cndp1*-KO mice, H_2_S formation was lower compared to female non-diabetic KO littermates, presumably due to the 3-fold lower kidney anserine and carnosine concentrations.

Our findings do not exclude beneficial effects of pharmacological doses of anserine and carnosine in diabetes via induction of H_2_S. Numerous studies in rodents demonstrated kidney protection in diabetic mice by carnosine administration [5,6,7,56,57]. Only two studies reported on the administration of anserine [15,58], even though nephroprotective effects of carnosine may at least in part be exerted by methylation to anserine [59]. We recently demonstrated several-fold increased kidney anserine concentrations after oral carnosine supplementation in healthy and db/db mice for four weeks [60]. None of these studies evaluated the interaction of therapeutic anserine or carnosine supplementation with H_2_S metabolism in diabetic mice, but in view of our finding, it deserves validation in animal models and in clinical trials. Likewise, our findings are of interest with regard to the potential therapeutic impact of combined treatment such as in patients with ischemic heart disease treated with the H_2_S donor thiosulfate. Anserine may not only exert direct beneficial effects, but also increase H_2_S availability.

A limitation of our study is the measurement of H_2_S synthesis capacity, based on an endpoint quantification with Ag_2_S, providing a high specificity and reproducibility with reasonable sensitivity. This, however, does not provide information on actual cell and tissue H_2_S concentrations. Current methods to quantify H_2_S concentrations comprise colorimetric assays, gas chromatography, fluorescence probes and electrochemical technics, and the advantages and limitations have recently been summarized [61]. Future studies combining different methods may further increase insights and validity of our findings. In the same direction, we demonstrated the synthesis activity of CBS and CSE in murine kidney tissue, and the functional role of HSP70, but not the specific mode of action of HSP70 on enzyme expression and activity.

## 5. Conclusions

In conclusion, we demonstrate a novel mechanism of action of the histidine-containing dipeptides anserine and carnosine, i.e., the induction of H_2_S synthesis in proximal tubular epithelial cells and capillary, venous and aortic endothelial cells, which in endothelial cells is exerted via HSP70. The in vivo relevance of these interactions is demonstrated in *Hsp70*- and *Cndp1*-KO mice, but awaits clarification in diabetes mellitus and ischemic heart disease. The positive feedback interaction of H_2_S, inhibiting anserine and carnosine degradation by CN1 together with the H_2_S synthesis-inducing effect of both dipeptides, should enhance the efficacy of therapeutic interventions with anserine, carnosine and with H_2_S donors.

## Figures and Tables

**Figure 1 antioxidants-12-00066-f001:**
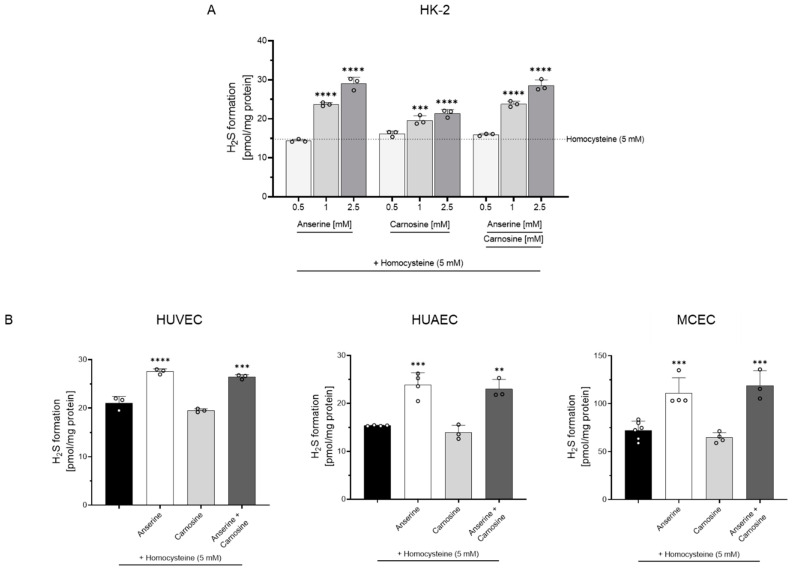
H_2_S formation in proximal tubular epithelial (HK-2) and endothelial cells (HUVEC, HUAEC, MCEC). (**A**) Anserine and carnosine dose-dependently increased H_2_S formation in HK-2 cells. (**B**) Treatment of HUVEC, HUAEC and MCEC with anserine but not carnosine (each 1 mM) increased H_2_S formation compared to untreated cells. **: *p* < 0.01; ***: *p* < 0.001; ****: *p* < 0.0001 (one-way ANOVA with Tukey’s test).

**Figure 2 antioxidants-12-00066-f002:**
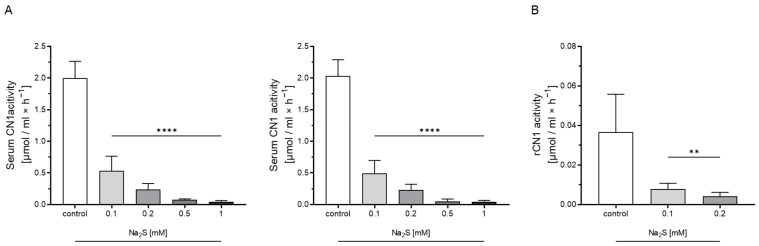
Effect of H_2_S on human serum CN1 activity and on recombinant CN1 activity (rCN1). (**A**) Addition of H_2_S donor sodium disulfide (Na_2_S) reduced human serum CN1 activity dose-dependently for carnosine (left panel) and anserine (right panel) as the substrate. (**B**) Degradation rate for carnosine was also reduced by Na_2_S for the recombinant carnosinase 1 (rCN1). **: *p* < 0.01; ****: *p* < 0.0001 (one-way ANOVA with Tukey’s test).

**Figure 3 antioxidants-12-00066-f003:**
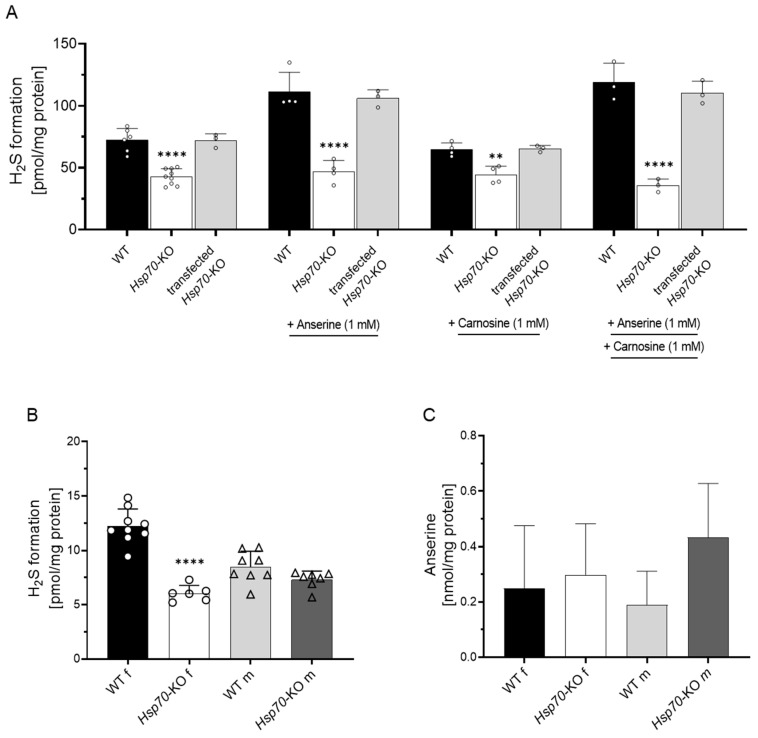
Decreased H_2_S formation in *Hsp70* knockout MCEC and mice. (**A**) H_2_S formation was lower in *Hsp70*-KO MCEC compared to WT MCEC; *Hsp70*-transfection restored H_2_S formation. (**B**) Renal H_2_S formation was lower in female *Hsp70*-KO compared to female WT mice. (**C**) Renal anserine concentrations were similar in all four groups. **: *p* < 0.01; ****: *p* < 0.0001 (one-way ANOVA with Tukey’s test).

**Figure 4 antioxidants-12-00066-f004:**
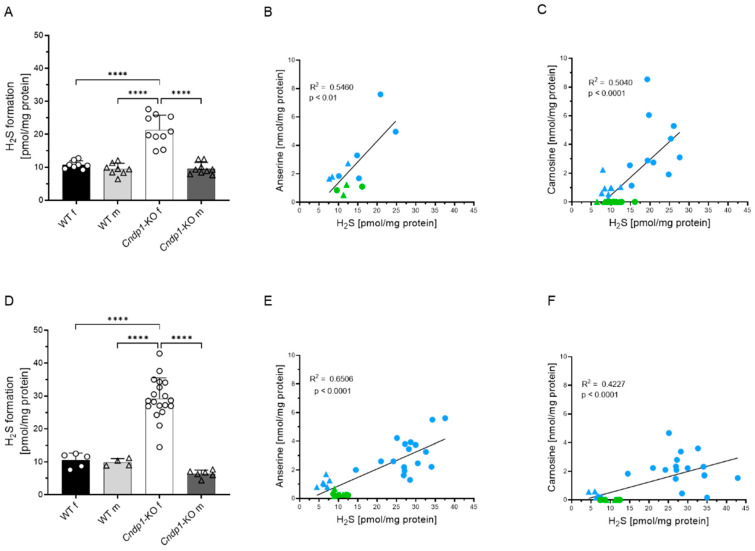
Renal H_2_S formation in WT and *Cndp1*-KO mic**e.** (**A**–**C**) H_2_S formation in kidney tissue of 23-to 25-week-old and (**D**–**F**) 47–51-week-old mice. H_2_S formation was increased in female *Cndp1*-KO vs. female control mice. Kidney H_2_S formation correlated with renal anserine concentrations (**B**,**E**) and with renal carnosine concentrations (**C**,**F**) in *Cndp1*-KO (green symbols) and WT (blue symbols) mice. Circles depict female and triangle male mice. ****: *p* < 0.0001 (one-way ANOVA with Tukey’s test).

**Figure 5 antioxidants-12-00066-f005:**
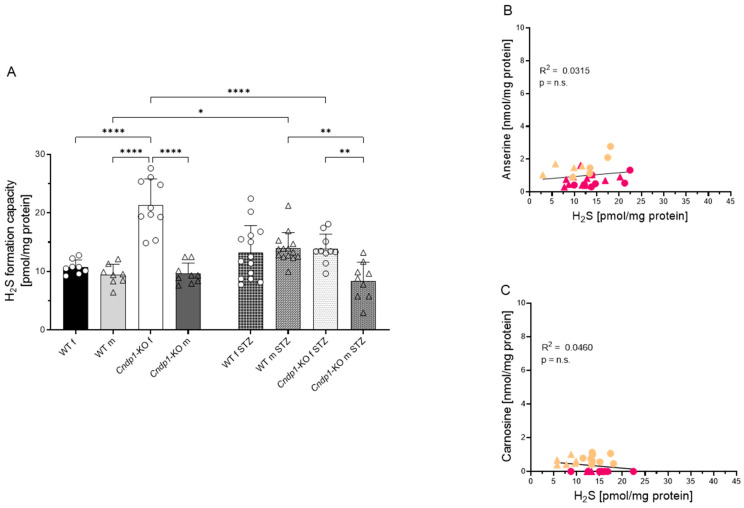
Renal H_2_S formation in diabetic WT and *Cndp1*-KO mice. (**A**) In female diabetic *Cndp1*-KO mice, kidney H_2_S formation was reduced compared to non-diabetic female *Cndp1*-KO mice and similar to diabetic WT mice. In male diabetic *Cndp1*-KO mice, kidney H_2_S formation was even reduced compared to male diabetic WT mice. (**B**) Kidney anserine and (**C**) carnosine concentrations were markedly reduced in diabetic *Cndp1*-KO mice (beige symbols) compared to diabetic controls (pink symbols) and not related with H_2_S formation. Circles depict female and triangle male mice. *: *p* < 0.05; **: *p* < 0.01; ****: *p* < 0.0001 (one-way ANOVA with Tukey’s test).

**Figure 6 antioxidants-12-00066-f006:**
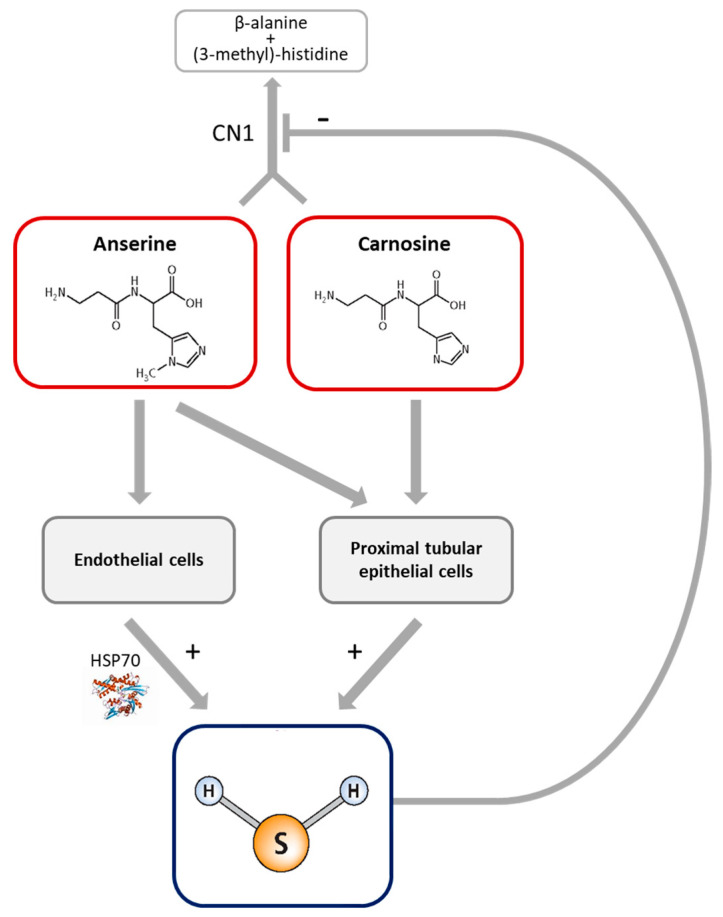
Anserine and carnosine induce H_2_S formation in a cell type- and Hsp70-specific manner within a positive feedback loop with CN1.

**Table 1 antioxidants-12-00066-t001:** Renal anserine and carnosine concentrations in kidney tissue of WT and *Cndp1*-KO mice (n = 3–17).

9	Age of Mice (Weeks)	Gender	Anserine (nmol/mg)	Carnosine (nmol/mg)	H_2_S Formation (pmol/mg)
WT	23–25	f	0.8 ± 0.4	<0.1 nmol	10.74 ± 1.21
WT	23–25	m	0.7 ± 0.5	<0.1 nmol	9.44 ± 1.78
*Cndp1*-KO	23–25	f	3.0 ± 1.3	3.8 ± 2.36	21.33 ± 4.46 ^a^
*Cndp1*-KO	23–25	m	1.8 ± 0.8 ^a^	0.9 ± 0.47	9.62 ± 1.79
WT	47–51	f	0.3 ± 0.04	<0.1 nmol	10.48 ± 2.20
WT	47–51	m	0.3 ± 0.2	<0.1 nmol	9.88 ± 1.11
*Cndp1*-KO	47–51	f	3.1 ± 1.3 ^a^	1.81 ± 0.72	29.22 ± 6.24 ^a^
*Cndp1*-KO	47–51	m	1.1 ± 0.3 ^a^	0.32 ± 0.19	6.38 ± 1.14 ^a^
STZ WT	23–25	f	0.7 ± 0.4	<0.1 nmol	13.20 ± 4.62
STZ WT	23–25	m	0.8 ± 0.4	<0.1 nmol	13.43 ± 1.67 ^b^
STZ *Cndp1*-KO	23–25	f	1.0 ± 0.4 ^b^	0.80 ± 0.29 ^b^	13.84 ± 2.53 ^b^
STZ *Cndp1*-KO	23–25	m	1.2 ± 0.5 ^a,b^	0.68 ± 0.31	8.38 ± 3.21 ^a^

^a^ = significantly different compared to respective WT group, ^b^ = significantly different compared to respective non-diabetic group. Statistical analysis was done by unpaired Student’s *t*-test.

## Data Availability

The data are contained within the article and Appendix A.

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
