# Peer review of "Anserine and Carnosine Induce HSP70-Dependent H2S Formation in Endothelial Cells and Murine Kidney"

_antioxidants, 2022, doi:10.3390/antiox12010066_

Round 1
Reviewer 1 Report
Review for manuscript entitled “Anserine and carnosine induce HSP70-dependent H2S-
formation in endothelial cells and mice kidney” by Wetzel et al.
The work investigates the interactions between nephroprotective agents anserine and carnosine and the gasotransmitter H2S in cell types (epithelial and endothelial cells) and animal models relevant for the assessment of kidney function and dysfunction. The paper investigates the relationship of anserine, carnosine and H2S in experimental diabetes induced by streptozocin, a pathology that can lead to deterioration of kidney function. The results lead to the conclusion that anserine and carnosine induce H2S formation in a cell-type and Hsp70-specific manner. Mechanistically, the interrelation of anserine and carnosine metabolism with H2S and Hsp70 is suggested to be part of a positive feedback loop with the enzyme carnosinase-1.
The manuscript is well-written and the results are presented with clarity. The topic suits the scope and broad audience of Antioxidants. The novelty of findings and the careful experimental execution merit its consideration for publication.
I have the following comments for the authors:
1. Title. The word “mice” could be replaced by the word “murine”.
2. Abstract. The abstract describes what was studied and the very rich experimental models that were employed. A sentence stating what was the central hypothesis will be very helpful to position the readership on what is the central goal of the investigation.
3. Abstract. The gasotransmitter H2S is described as being “cytoprotective”. As it is the case with other gasotransmitters, the actions of H2S can be protective as well as damaging to cells and tissues depending on the concentration. Please, rewrite this to depict this dual nature of H2S, or else, remove the word “cytoprotective”.
4. Results 3.3. and 3.4. The authors identified reduced H2S production in the kidney biopsies of female mice Hsp70 knock-out, but this was not the case in male mice. Chronic elevation of anserine and carnosine caused by mutations in carnosinase 1 led to marked elevations of H2S in both female and male mice, albeit, the effects were more marked in female mice. What is the possible reason for these sex-dependent effects in short-term versus chronic exposure to anserine/carnosine? Is there any known sex-specific regulation of anserine, carnosine or H2S metabolism?
5. Figures 4 and 5: the colouring of blue and green dots in graphs presented in Figures 4 and 5 represent different things, and makes it hard to dissect when looking at the two figures together for context. In Figure 4, green and blue dots represent Cndp1-KO and WT mice, respectively. In Figure 5, green and blue dots represent diabetic Cndp1-KO mice and non-diabetic controls, respectively, i.e. both in the Cndp1-KO background (not WT). A single color could be used for Cndp1-KO under control, normal conditions, in both figures. But then, WT and diabetic Cndp1-KO should each have a different color.
6. Discussion, line 278. Both anserine/carnosine and H2S act via activation of HSP70 [14, 15, 19, 34]. This statement is correct, however, incomplete. The actions of H2S involve a variety of molecular mechanisms, which are themselves a matter of intense debate, as there is no single specific receptor for this gasotransmitter. This has to be specified. Actions of H2S via Hsp70 have been documented, however, so are many other effects of H2S not involving Hsp70.
7. The study shows that H2S reduces CN1 activity. Previous work by the authors demonstrated inhibition of CN1 by thiols, including Cys. The mode of inhibition appeared to be specific toward thiolation of Cys102 of CN1, which is a catalytic amino acid residue. Would a similar mechanism be expected for the reduction of CN1 activity by H2S? Do the authors have any evidence with recombinant CN1 that a direct reaction of H2S with Cys102 occurs, i.e. that this may be the molecular mechanism underlying the effect of the gasotransmitter on CN1 activity?
8. Anserine stimulates H2S production via a mechanism that involves Hsp70. Is there any evidence that Hsp70 either stabilizes or stimulates the activity of H2S- producing enzymes?
9. Do patients with inherited diseases in anserine and carnosine metabolism exhibit abnormal concentrations of H2S in plasma, cells and tissues?
If points 5 to 8 are not known, a discussion on these points can be included as a new section toward the end of the manuscript, elaborating on “Limitations and further studies”.
Minor comments
Line 202. Replace informal didn’t for did not.
Line 218. Replace extend for extent.
Supplemental data. please, revise and correct spelling and typos, for example, western blot is two words.
Figure S3. The size of the frames of the western blot images does not match that of the actual membranes containing the experimental results. As presented, there are indications of molecular weight marks (arrows) that fall outside the “real membrane”. Please, correct a necessary or provide an unadjusted version of the western blot membranes.
Figure S3, legend. Please, correct typos, for example, “vom left to right”.
Figure S4, legend. These graphs do not show any “rates” of H2S formation; i.e. these are not time-resolved measurements. What seems to the different is the yield, or cumulative end-point H2S production.
Figure S5, legend. Please, specify what type of biological sample is being measured, i.e. kidney biospies?
Reviewer 2 Report
The current study by Wetzel et al. attempts to demonstrate Anserine and carnosine-induced H2S formation and the role of Cndp1 and Hsp70 in endothelial cells and kidney epithelial cells. The reviewer has the following observations:
1. The authors haven’t determined the H2S levels in the manuscript. The assay used in the manuscript is the enzyme production using certain substrates rather than the levels of sulfide metabolites. This should be determined using standard methods from the literature.
2. The authors don’t determine the expressions of H2S-producing enzymes, including CBS, CSE, and MST to see how these enzymes are involved in regulating HSP70-dependent H2S formation mediated by Anserine and carnosine. Additionally how and where Cndp1 is involved in this process is not clearly demonstrated.
3. The authors have not determined cell toxicity or lack of from Anserine and carnosine have not been determined. Moreover, the use of these concentrations has not been justified.
4. The authors determine that the H2S reduces human serum CN1 activity and on recombinant CN1 activity (rCN1); however, this is not the effect of CN1 release rather than the modification of CN1 by H2S possibly its kinetics and H2S mediated protein modification. This has not been clearly demonstrated. The possibility or lack of H2S-mediated posttranslational modification such as sulfhydration/persulfidation.
5. Additionally, the H2S concentration used was relatively higher than the literature and can also be toxic for endothelial cells. H2S can have negative implications on molecular signaling under high concentrations.
6. There is no demonstration of kidney function in the whole study and how it correlates with the Anserine, carnosine, and Hsp70. Although the role of H2S has been known in the literature on kidney function, the authors do not demonstrate this in these models.
7. The authors add complexity to the experimental design, which lacks clarity and a linear approach in proof of concept. The presence of STZ can reduce H2S but how this is delineated and have any relevance in the current study is not properly discussed. Additionally, podocytes play a crucial role in kidney function and this has been largely ignored apart from the lack of kidney function and associated biochemistry.
8. The authors mention the use of homocysteine and changes in H2S levels in the supplementary figure. This makes the reviewer think that the authors lack a sound understanding of the H2S signaling and the biological implications of increased homocysteine levels.
Reviewer 3 Report
This is an interesting study where the authors have shown that the nephroprotective effects of anserine and carnosine (histidine-containing dipeptides) are mediated through hydrogen sulfide formation. The y use in vitro and carnosinase-1 knock-out (Cndp1-KO) and Hsp70-KO mice, as well as type 1 diabetic mice.
Overall, this is a well conducted study. However, there are some gaps in presentation of data, which the authors need to address.
1. Diabetes was induced by intraperitoneal 5 to 6 injections of Streptozocin (50mg/kg) at age 9 to 10 weeks. Please provide ketone body values in the urine.
2. Table 1 clearly shows differences in anserine, carnosine, and hydrogen sulfide levels that are gender dependent in Cndp1-KO mice. These differences need to be highlighted in the abstract.
3, Although the effects of anserine and carnosine are attributed to modulation of hydrogen sulfide levels, please provide some data on the actions of these dipeptides on serum and kidney tissue levels of nitric oxide levels.
Round 2
Reviewer 2 Report
In this revised submission by Wetzel et al., although the authors have acknowledged the study's limitations and revised some parts of the manuscript, most of the reviewer's questions remain unsatisfactory.
What type of H2S has been produced? This may also include thiosulfate apart from other sulfide species. Whether they're oxidized or in what form has not been differentiated. Moreover, the H2S formation can be multifactorial that includes many substrates and enzymes in the process, which hasn't been clearly demonstrated.
Moreover, the authors claim they've demonstrated CSE and CBS activity, which is not mentioned in the methods nor can the reviewer see in the manuscript. Which enzyme is involved and why is still not demonstrated convincingly.
Even if considered that Anserine and carnosine induce H2S formation, how does it matter? How it is relevant for kidney function is not demonstrated.
